# Transcriptomic Analysis of Skin Tissue Reveals Molecular Mechanisms of Thermal Adaptation in Cold-Exposed Lambs

**DOI:** 10.3390/ani15101405

**Published:** 2025-05-13

**Authors:** Mengyu Feng, Kaixi Ji, Yutao Li, Pâmela Almeida Alexandre, Dan Jiao, Yanping Liang, Xia Du, Xindong Cheng, Huitong Zhou, Jon G. H. Hickford, Guo Yang

**Affiliations:** 1State Key Laboratory of Ecological Safety and Sustainable Development in Arid Lands, Northwest Institute of Eco-Environment and Resources, Chinese Academy of Sciences, Lanzhou 730000, China; fengmengyu@nieer.ac.cn (M.F.); jikaixi@nieer.ac.cn (K.J.); jiaodan@lzb.ac.cn (D.J.); liangyp1045@163.com (Y.L.); dudxyy@163.com (X.D.); chengxindong@126.com (X.C.); 2University of Chinese Academy of Sciences, Beijing 100049, China; 3Institute of Animal Husbandry and Veterinary Medicine, Shandong Academy of Agricultural Sciences/Shandong Key Laboratory of Animal Microecologics and Efficient Breeding of Livestock and Poultry, Jinan 250100, China; 4CSIRO Agriculture and Food, Brisbane 4067, Australia; yutao_li@hotmail.com (Y.L.); pamela.alexadre@csiro.au (P.A.A.); 5Gene-Marker Laboratory, Department of Agricultural Sciences, Lincoln University, Lincoln 7647, New Zealand; huitong.zhou@lincoln.ac.nz (H.Z.); jon.hickford@lincoln.ac.nz (J.G.H.H.); 6Yellow River Estuary Tan Sheep Institute of Industrial Technology, Dongying 257400, China; 7Shandong Huakun Rural Revitalization Institute, Jinan 250014, China

**Keywords:** transcriptome, Hulunbuir lambs, Hu lambs, cold, skin, wool

## Abstract

The ruminant livestock farming regions of China are primarily located in the North, where the climate is characterized by seasonal sub-zero temperatures and large temperature fluctuations that can lead to cold stress. Hair fibers and skin serve as primary insulative layers, playing a crucial role in maintaining body temperature in cold environments, so this study aimed to explore the mechanisms of wool and skin response to a cold challenge in lambs of two breeds of indigenous Chinese sheep, the Hulunbuir and the Hu. Several physiological changes in response to the challenge were measured, with similar patterns observed in both the Hulunbuir and Hu lambs, although the changes were more pronounced in the former breed. At −20 °C, both breeds activated pathways related to the immune and endocrine systems, signal transduction, development, and regeneration. The TNF signaling pathway and osteoclast differentiation exhibited enrichment not only in Hulunbuir-specific differentially expressed genes (DEGs) but also in DEGs shared between breeds. These findings suggest their potential as key regulatory pathways mediating cold adaptation mechanisms in lambs, with implications for improving animal welfare standards and fortifying livestock industry resilience against global climatic perturbations.

## 1. Introduction

The climate in Northern China is characterized by seasonal sub-zero temperatures and large temperature fluctuations. These present a challenge to animals and animal husbandry. Cold tolerance refers to the ability of animals to withstand low ambient temperatures, and Slee et al. [1] observed that coat characteristics, skin thickness, and birthweight were strongly and positively correlated with cold tolerance in newborn Merino lambs.

Hair fibers and skin act as a primary insulative layer and thus play a role in maintaining temperature in colder environments [2,3]. Subcutaneous fat also serves as an insulative layer, with a thicker fat layer helping animals maintain body temperature and reduce their heat loss [4]. It is notable though that newborn lambs do not possess an insulative subcutaneous fat layer [5].

In sheep exposed to lower temperatures, wool plays an important role in maintaining the core temperature [6], and environmental temperature responses can be categorized into the following zones: cold stress, thermoneutral, and heat stress [7]. The thermoneutral zone is the range of ambient temperatures in which the metabolic rate is at a minimum and is influenced by various factors, including breed, age, level of nutrition, and wool cover. Before shearing, the minimum critical temperature for sheep is −4 °C [8], but after shearing, this temperature may rise to 28 °C [9]. Studies indicate that the minimum critical temperature also varies with different wool lengths and nutrition [10].

Skin and coat traits also affect heat tolerance in cattle [11]. Research has revealed that thin and smooth skin is associated with better heat tolerance. Moreover, sheep with longer hairs and thicker coats tend to exhibit higher sweating rates, respiratory rates, and skin temperatures, which can be a threat to survival in environments where temperatures exceed 40.5 °C [12]. Environmental factors also cause changes in animal skin and coat traits. For example, low temperatures in colder climates promote hair growth, whereas high humidity in tropical regions inhibits it [13]. Additionally, different allele frequencies in genes related to hair follicle development exist between goat breeds from Northern and Southern China [14], and Yaks and Tibetan sheep exhibit adaptations suited to extreme environments, such as having thicker hair during colder winters [15,16].

Hulunbuir sheep inhabit the colder regions of Northern China and are hardy animals with well-developed muscularity [17]. In contrast, Hu sheep originate from the warmer and more humid regions of Southern China, and they are known for their white lambskin, narrower body shape, and relatively slender limbs [18]. Both breeds have white and heterogeneous fleeces that include primary and secondary hair follicles. The primary hair follicles produce medullated wool, which can be further categorized into coarse wool, heterotypical hair, and kemp, while the secondary hair follicles produce wool, with this ‘under-fleece’ typically consisting of non-medullated fibers [17,18].

Despite the importance of wool, skin, and fat in sheep survival in colder environments, the mechanisms underlying these adaptations remain unclear. Accordingly, based on the different genetics and selection of Hulunbuir and Hu lambs, we hypothesize that they possess both common and unique molecular mechanisms for increasing resilience in colder environments. The aim of this study was therefore to compare the skin transcriptome of these two breeds, as well as selected physiological traits and wool and skin characteristics, to better understand the molecular mechanisms responsible for cold tolerance. We expected that breed differences would exist.

## 2. Materials and Methods

The Animal Welfare and Experimental Ethics Committee of the Northwest Institute of Ecological Environment and Resources, Chinese Academy of Sciences, approved all experimental procedures (approval number CAS201810082), which complied with People’s Republic of China standard GB/T 35892-2018 for Laboratory Animal Welfare [19] and meeting ARRIVE (Animal Research: Reporting of In Vivo Experiments) 2.0 guidelines.

### 2.1. Location, Animals, and Their Management

Recently shorn lambs (*n* = 20; shorn to an equivalent wool length) obtained from the Darhan-Muminggan Joint County Farm in inner Mongolia and of similar age (8 months), belonging to the Hulunbuir (*n* = 10; average 34.5 ± 0.70 kg) and Hu (*n* = 10; average 34.9 ± 0.79 kg) breeds, were raised at the Ecological and Agricultural Experimental Station, Gaolan, Gansu Province, People’s Republic of China (36°13″ N, 103° 47″ E), at an altitude of 1780 m above sea level. The lambs from each breed were divided into two groups that were matched for body weight and housed in individual metabolic cages (1.2 m × 0.6 m × 1.8 m) in temperature-controlled rooms with a daily 12 h light and 12 h dark cycle.

Throughout the experiment, lambs had free access to water and were fed twice daily at 0830 and 1830. Each lamb received a daily ration of 1200 g in total, with a diet that was 70% roughage and 30% concentrate. The ratio aligns with previous studies that recommends balanced roughage-to-concentrate ratios for fattening lambs to achieve optimal growth performance and carcass quality under controlled conditions [20]. The 30% concentrate ensured sufficient metabolizable energy intake (11.0 MJ/d) and crude protein (164.2 g/d) to meet the growth demands of lambs under cold stress, supporting a target daily weight gain of 100 g. The 70% roughage maintained adequate neutral detergent fiber (NDF) and acid detergent fiber (ADF) levels to sustain rumen motility, pH stability, and microbial activity. The fixed ratio minimized variability in nutrient intake between the lambs, ensuring that the differences measured between them could be attributed to cold stress rather than dietary fluctuation, and the composition and nutritional content of the diet are shown in Table 1.

### 2.2. Experimental Design and Treatments

One group of Hulunbuir lambs was kept at 15 °C (HB+15; *n* = 5), and the other group (HB−20; *n* = 5) was exposed to −20 °C in temperature-controlled rooms. Similarly, the Hu lambs were divided into one group at 15 °C (HU+15; *n* = 5), and the other group was exposed to −20 °C (HU−20; *n* = 5). For the cold-exposed groups, the lambs were first acclimatized at 15 °C for one week; then, the temperature was decreased from 15 °C to −20 °C over 14 days by intervals of 5 °C every 2 days. After reaching −20 °C, this temperature was maintained for 38 days.

### 2.3. Sample Collection and Physiological Response Variables

On the first day of temperature reduction, the wool on the right mid-side of each lamb was dyed. On day 52, wool samples were collected from both the left and right mid-sides of each lamb and placed in separate sealed bags.

The respiratory rate of the lambs was observed by counting the number of chest wall movements per minute while the lambs were in a calm state. Each lamb was measured three times, and the average of the three measurements was taken as the respiratory rate for that lamb. Next, the rectal temperature of the lamb was measured using an electronic thermometer (Omron MC-347). Rectal temperature measurements were conducted at 0700 (pre-feeding) to minimize confounding effects of circadian rhythm or postprandial metabolic changes.

The lambs were fasted for 12 h prior to slaughter, and their final body weight was measured and recorded. After anesthesia, the lambs were then exsanguinated by cutting the carotid artery, and the hot carcass weight (carcass minus head, skin, and gut/organs) was recorded. Skin samples (approximately 2 cm × 2 cm) were harvested from the left-side region of each lamb and stored in 4% paraformaldehyde. Additional smaller skin samples (approximately 1 cm × 1 cm) were also collected from the same left-side region, frozen in liquid nitrogen, and stored at −80 °C for further analysis [21].

### 2.4. Wool and Skin Response Variables

Wool on the left mid-side of the lambs was dyed at the start of the experiment. At the end of the experiment, wool from the center of the dyed patch was cut from the base, and the length of undyed wool was measured to determine wool length growth. The coarse and fine wool fibers were separated, and the diameter of both fiber types was measured under a projection microscope to assess mean fiber diameter [22]. Wool grease content was determined using a Soxhlet extractor (Foss, Copenhagen, Denmark), following the established procedure described by Pu et al. [23]. Separate wool samples (approximately 2 cm × 2 cm) were collected from the same location on the mid-side of each of the lamb to determine the proportion of fine wool fibers in the total collected fiber sample. The weight of wool sample was regarded as wool yield.

The paraformaldehyde-fixed skin samples were dehydrated in 100% ethanol, cleaned in xylene for 2 h, and then embedded in paraffin. Five-micron transverse and vertical sections were cut, mounted on slides, and stained with hematoxylin and eosin. Hair follicle density and epidermis thickness were measured. Ten different microscopic fields were examined to determine the mean hair follicle density for each skin sample [24].

### 2.5. Transcriptome Sequencing and Differentially Expressed Gene Analyses

A total of 20 skin samples from Hulunbuir and Hu lambs (5 samples per group) were selected for RNA analysis. Total RNA was extracted using the Trizol reagent (Invitrogen, Carlsbad, CA, USA), and the RNA integrity, concentration, and purity were assessed using an Agilent 2100 Bioanalyzer (Agilent, Santa Clara, CA, USA). Poly-A messenger RNA (mRNA) was isolated using poly-T oligo-attached magnetic beads, and first-strand cDNA was synthesized using random primers and reverse transcriptase (Super SCRIPT II Reverse transcriptase, Invitrogen, Carlsbad, CA, USA). Second-strand cDNA was synthesized using DNA polymerase I. Purified double-stranded cDNA was repaired and ligated to connectors, then PCR-amplified, and quality-assessed using the Agilent 2100 Bioanalyzer (Agilent). The resulting cDNA library was sequenced on a BGISEQ-500 platform (BGI, Shenzhen, China).

The raw sequence reads were quality-checked, trimmed, and aligned to the Ovis aries reference genome (Oar_v4.0; GCF_000298735.2) [25] using the Seqera Nextflow v23.10.1 RNA-seq pipeline (nf-core/rnaseq v3.14.0 (https://nf-co.re/ (accessed on 27 May 2024)). In brief, raw read quality control was assessed using FastQC v0.12.1; adapter and any low-quality reads were removed with TrimGalore v0.6.7. Trimmed FastQ files were mapped to the genome using STAR v2.7.10a, and quantification (raw counts per gene) was performed using RSEM v1.3.1. All tools’ versions and full details of the pipeline are available at https://nf-co.re/rnaseq/ (accessed on 27 May 2024). Gene expression levels were normalized across all samples using edgeR v4.2.1 in R v4.3.3. The raw read counts were transformed to log2 counts per million (CPM), minimally expressed genes (CPM < 1 in 5 samples) were filtered out, and libraries were normalized by the trimmed mean of M-values approach [26]. After normalization, the genes with at least 1 CPM in at least 5 samples were retained for differential expression analysis, which was undertaken for temperature and breed comparisons using negative binomial generalized linear models. The False Discovery Rate (FDR) was calculated by *p*-value to control for multiple comparisons. This approach involves adjusting the *p*-values to account for the likelihood of false positives that occur when conducting multiple simultaneous tests. The FDR calculation was performed using the Benjamini–Hochberg procedure, which ranks the *p*-values in ascending order and determines the threshold at which the expected proportion of false discoveries is controlled [27]. A principal component analysis was conducted using the autoplot function in R v4.3.3, and differentially expressed genes (DEGs) were identified based on having an FDR < 0.05.

### 2.6. Functional Enrichment and Co-Expression Module Analysis

Gene Ontology (GO) and KEGG pathway enrichment analyses of the DEGs were performed using KOBAS v3.0 (http://bioinfo.org/kobas; accessed on 2 July 2024)), with GO terms and pathways considered enriched at *p* < 0.05. Since the KOBAS website only accepts enrichment analysis for fewer than 3000 genes, we filtered DEGs for comparison groups with a high number of DEGs, including the temperature comparison group (HB+15 vs. HB−20) and unique DEGs in HB+15 vs. HB−20. We selected DEGs that met the criteria of FDR < 0.05 and |Log2FoldChange| ≥ 1 for Gene Ontology (GO) terms and pathway enrichment analyses. Additionally, we constructed networks for the common DEGs and pathways related to colder environments, as well as the unique DEGs and pathways in the HU−20 lambs using Cytoscape v3.10.2 software (https://cytoscape.org; accessed on 14 August 2024). For the gene co-expression network analysis, all the DEGs and the 14 characteristics were used as nodes, and connections between them were identified using the partial correlation and information theory algorithm. To explore the relationship between genes and the variables in colder environments, we built cold-specific networks across all breeds and visualized these using Cytoscape.

### 2.7. Validation with Reverse Transcription-Quantitative PCR (RT-qPCR)

To validate the results of transcriptome analyses, four DEGs from each of the four groups were selected for analysis using an RT-qPCR approach, with actin beta gene (*ACTB*) as an internal reference. Primers for this analysis were designed using Primer 5.0 software (PREMIER Biosoft International, Palo Alto, CA, USA; Table 2).

Total RNA samples were reverse-transcribed using the PrimerScript RT reagent kit with gDNA Erase (Servicebio, Wuhan, China) following the manufacturer’s instructions. Next, RT-qPCR was performed with an Agilent Mx3000P system (Agilent) and Universal Blue SYBR Green qPCR Master Mix (Servicebio, Wuhan, China). The amplification protocol consisted of an initial denaturation step at 95 °C for 30 s, followed by 40 cycles of denaturation at 95 °C for 30 s and annealing/extension at 60 °C for 30 s. The relative expression levels were analyzed using the 2^−ΔΔCt^ method [28].

### 2.8. Statistical Analyses

The various characteristics (respiratory rate, rectal temperature, average daily gain, hot carcass weight, wool growth length, mean fiber diameter, wool grease content, the proportion of fine wool fibers, wool yield, hair follicle density, and epidermis thickness) were analyzed using SPSS 27.0 (IBM-SPSS Inc., Chicago, IL, USA). A two-way ANOVA framework was implemented through the General Linear Model (GLM), where respiratory rate served as the dependent variable, with temperature and breed designated as fixed factors. Prior to hypothesis testing, diagnostic checks, including descriptive statistics and homogeneity of variance assessments, were systematically conducted to validate model assumptions. Upon confirming assumption compliance, post hoc pairwise comparisons among the four groups were executed via the least significant difference (LSD) method to delineate specific subgroup disparities. All data were visualized using GraphPad Prism 10.2.3 (GraphPad Software Inc., San Diego, CA, USA). Significance was defined at *p* < 0.05, with all results reported as means ± standard error of the mean (SEM) to ensure precision in effect size interpretation.

## 3. Results

### 3.1. Changes in Physiological Traits, Wool, and Skin Characteristics

For both the Hulunbuir and Hu lambs, the temperature they were maintained at influenced average daily gain, rectal temperature, respiratory rate, wool growth length, hair follicle density, and epidermis thickness.

The ANOVAs (Table 3 and Appendix A; Appendix A) comparing wool and skin characteristics and physiological traits at the two temperatures (−20 °C vs. 15 °C) revealed that both breed and temperature and the interaction of these two variables affected selected characteristics.

Temperature affected wool growth length (*p* < 0.001), but breed did not, although breed and temperature interacted with this characteristic (*p* = 0.046). While breed affected the fine wool mean fiber diameter and wool yield (*p* = 0.004 and *p* = 0.037, respectively), neither trait was affected by the temperature. Temperature affected hair follicle density (*p* = 0.044; Figure 1A), epidermis thickness (*p* = 0.002; Figure 1B), average daily gain (*p* = 0.037), rectal temperature (*p* < 0.001), and respiratory rate (*p* = 0.001).

The LSD post hoc comparisons (Table 4 and Appendix A; Appendix A) revealed that, at −20 °C, wool growth length was greater in both breeds compared to their breed contemporaries at 15 °C (Hulunbuir: *p* < 0.001; and Hu: *p* = 0.028). There was no difference in wool growth length between the breeds at −20 °C or at 15 °C.

The fine wool mean fiber diameter differed between the Hu and Hulunbuir lambs at 15 °C (*p* = 0.014), and the wool yield differed (*p* = 0.025) between the Hu and Hulunbuir lambs at −20 °C. The hair follicle density and epidermis thickness differed between the Hulunbuir lambs at 15 °C and those at −20 °C (*p* = 0.041 and *p* = 0.006, respectively), but this effect was not observed with the Hu lambs. Average daily gain and hot carcass weight differed for the Hu lambs at 15 °C compared to those at −20 °C (*p* = 0.042 and *p* = 0.030, respectively). Rectal temperatures differed for both breeds when comparing them at 15 °C and −20° (Hu: *p* = 0.004; and Hulunbuir: *p* < 0.001), but the respiratory rate only differed in the Hulunbuir sheep at the two different temperatures (*p* < 0.001).

### 3.2. Quality Assessment of RNA-Seq Data and Genome Alignment

A total of 882.2 million clean reads were obtained with a single-base sequencing error rate of less than 1%. Each sample contained between 45.2 million to 48.2 million reads. After quality control adjustment, the average proportion of sequence data that matched the reference genome was 94.37%. The average GC base content of the transcripts was 49.9%, suggesting that high-quality sequencing data were produced (Appendix A) and that the RNA-Seq data could validly be used for analysis.

### 3.3. Gene Expression Level and Cluster Analysis

#### 3.3.1. Temperature Contrasts Across Breeds and Breed Contrasts Across Temperatures

A principal component analysis revealed an effect of temperature on the skin transcriptome. This was observed as two distinct temperature clusters when comparing all the lambs at the two temperatures. There was no clear difference between the breeds (Figure 2A). The across-breed gene expression analysis that compared all the lambs that were at −20 °C with those at 15 °C revealed 3679 DEGs, 1657 that were upregulated, and 2022 that were downregulated (Figure 2B). The top five most upregulated genes were *FOSB*, *NR4A1*, *FAM71A*, *LOC101113168,* and *FOS*. In contrast, only five DEGs were detected when contrasting the two breeds across temperatures, all of which were upregulated in the Hulunbuir lambs (Figure 2B).

#### 3.3.2. Temperature Contrast Within Breed

A total of 2724 DEGs were identified when the HB+15 group was compared with the HB−20 group, with 1397 upregulated and 1327 downregulated (Figure 2B). The top five upregulated genes in this within-breed comparison were *FOSB*, *NR4A1*, *GNLY*, *LOC105613350,* and *FAM71A*. When the HU+15 group was compared with the HU−20 group, 122 DEGs were detected, with 52 upregulated and 70 downregulated (Figure 2B). The top five upregulated genes were *FOSB*, *NR4A1*, *FAM71A*, *APOLD1,* and *ATF3*.

#### 3.3.3. Breed Contrast Within the Same Temperature

When comparing different breeds at the same temperature, no DEGs were found in comparing the HU−20 group with the HB−20 group, but the comparison between the HU+15 and HB+15 groups revealed 81 DEGs, with 64 upregulated and 17 downregulated (Figure 2B). Only two DEGs were identified in both the temperature and breed comparison, *TSPAN13* and *MSRA*. These were upregulated in the Hulunbuir lambs compared to the Hu lambs but downregulated at −20 °C compared to 15 °C (Figure 3A). The Venn diagram of the DEGs in the comparisons between the various groups (Figure 3B) revealed 103 DEGs shared between HB+15 vs. HB−20 and HU+15 vs. HU−20, with 2555 DEGs unique to HB+15 vs. HB−20, and 19 DEGs unique to HU+15 vs. HU−20 at the two temperatures.

### 3.4. GO Analysis of the DEGs

Using the DEGs identified from the group comparisons, gene ontology (GO) analysis was performed. The DEGs were classified into three categories of function: biological process (BP), cellular component (CC), and molecular function (MF) (Figure 4). In the BP category, the DEGs from the temperature contrasts across breeds (i.e., HB−20 + HU−20 vs. HB+15 + HU+15), and the HU+15 vs. HU−20 groups were enriched in the ‘response to wounding’ category. With the CC category, the DEGs from both the temperature contrasts across breeds (i.e., HB−20 + HU−20 vs. HB+15 + HU+15) and the HU+15 vs. HU−20 groups were enriched in the ‘cytoplasm’ category. In the temperature contrast across breeds (i.e., HB−20 + HU−20 vs. HB+15 + HU+15), the DEGs were also enriched in the ‘mitochondrial inner membrane’ category, while, in the HU+15 vs. HU−20 groups, the DEGs were enriched in the ‘endoplasmic reticulum’ category. The DEGs from the HB+15 vs. HB−20 and HU+15 vs. HB+15 groups were enriched in the ‘cytosol’ category. For the MF category in temperature contrast across breeds (i.e., HB−20 + HU−20 vs. HB+15 + HU+15) and the HU+15 vs. HU−20 groups, the GO term related to ‘heme-binding’ was identified. For the HB+15 vs. HB-20 and HU+15 vs. HU−20 groups, the same 103 DEGs were enriched in the ‘response to wounding’ category in BP, ‘cytoplasm’ category and ‘endoplasmic reticulum’ category in CC, and ‘heme-binding’ in the MF category.

### 3.5. KEGG Analysis of DEGs

#### 3.5.1. Temperature Contrast Between Breeds

To further investigate the enrichment of DEGs in various signaling pathways, we used the KEGG database for pathway analysis, setting a statistical threshold of *p* < 0.05 to identify enriched pathways (Figure 5). In the temperature contrast between breeds (Figure 5A; HB−20 + HU−20 vs. HB+15 + HU+15), we identified 41 enriched pathways, including the IL-17 (oas04657) and TNF signaling pathways (oas04668), osteoclast differentiation (oas04380), apelin signaling pathway (oas04371), circadian entrainment (oas04713), aldosterone synthesis and secretion (oas04925), and osteoclast differentiation (oas04380). These pathways were mainly associated with organismal systems, environmental information processing, and metabolism.

#### 3.5.2. Temperature Contrast Within the Same Breed

In the HB+15 vs. HB−20 group comparison (Figure 5B), we identified 26 enriched pathways, including the TNF signaling pathway (oas04668), osteoclast differentiation (oas04380), the apelin signaling pathway (oas04371), the IL-17 signaling pathway (oas04657), parathyroid hormone synthesis, secretion and action (oas04928), and the oxytocin signaling pathway (oas04921). These pathways were primarily related to organismal systems, environmental information processing, and metabolism.

In the HU+15 vs. HU−20 comparison (Figure 5C), we identified 31 enriched pathways, including the IL-17 signaling pathway (oas04657), MAPK signaling pathway (oas04010), aldosterone synthesis and secretion (oas04925), and parathyroid hormone synthesis, secretion, and action (oas04928). These pathways were primarily associated with organismal systems, environmental information processing, genetic information processing, and cellular processes.

#### 3.5.3. Breed Contrasts at the Same Temperature

In the HB+15 vs. HU+15 comparison (Figure 5D), we identified 14 pathways, including fatty acid degradation (oas00071), protein export (oas03060), mineral absorption (oas04978), and valine, leucine, and isoleucine degradation (oas00280). These pathways are mainly related to metabolism, organismal systems, genetic information processing, and environmental information processing.

For the HB+15 vs. HB−20 and HU+15 vs. HU−20 groups (Figure 5E), the same 103 DEGs were enriched in 27 pathways, including the IL-17 signaling pathway (oas04657), the MAPK signaling pathway (oas04010), parathyroid hormone synthesis, secretion, and action (oas04928), TNF signaling pathway (oas04668), osteoclast differentiation (oas04380), oxytocin signaling pathway (oas04921), and aldosterone synthesis and secretion (oas04925). These pathways were mainly associated with organismal systems, environmental information processing, and genetic information processing.

In addition, in temperature contrast across breeds (i.e., HB−20 + HU−20 vs. HB+15 + HU+15), the HB+15 vs. HB−20 groups, the HU+15 vs. HU−20 groups and the HB+15 vs. HB−20 and HU+15 vs. HU−20 groups shared 11 enriched pathways, including the IL-17 signaling pathway (oas04657), parathyroid hormone synthesis, secretion, and action (oas04928), the TNF signaling pathway (oas04668), osteoclast differentiation (oas04380), the oxytocin signaling pathway (oas04921), and aldosterone synthesis and secretion (oas04925). These pathways were related to organismal systems and environmental information processing and hence may be key pathways enabling lambs to adapt to colder environments.

The 614 unique DEGs in the HB+15 vs. HB−20 comparison were enriched in 11 pathways, primarily including the apelin signaling pathway (oas04371), fluid shear stress and atherosclerosis (oas05418), the TNF signaling pathway (oas04668), osteoclast differentiation (oas04380), ABC transporters (oas02010), and taurine and hypotaurine metabolism (oas00430). These pathways were mainly associated with organismal systems, environmental information processing, and metabolism (Figure 5F).

### 3.6. Co-Expression Network Construction and Analysis

Based on the 11 common enriched pathways across the temperature-related comparison groups and the unique DEGs enriched in the Hulunbuir lambs at −20 °C, we constructed gene and pathway networks (Figure 6) to identify important pathways and genes related to colder environments in the skin. The genes *JUN*, *FOS*, *PTGS2*, *FOSB,* and *NR4A2* were associated with two or more pathways (Figure 6A), and the genes *SOCS3*, *NOS3*, *IL1A*, *JUNB*, *KLF2*, *CREB3L4*, and *NOX1* were similarly linked to two or more pathways (Figure 6B). 

Further combined analysis of these genes with the gene–phenotype network in colder environments revealed that *PTGS2* was associated with rectal temperature, while *NR4A2* was linked to the proportion of fine wool fibers (Figure 7). Among the unique DEGs in the HB+15 vs. HB−20 groups, *IL1A* was associated with wool yield, and *JUNB*, *CREB3L4*, *CACNA1E*, *RYR3*, *NDST3*, *OGT*, and *ABCA2* were related to the proportion of fine wool fibers (Figure 7). Most DEGs were related to the proportion of fine wool fibers, with *GNLY*, specific to the Hulunbuir lambs in colder environments, as the most significant. These DEGs were also associated with sheep traits that enable adaptation to colder environments.

### 3.7. Validation of RNA-Seq Data by RT-qPCR

The expressions of *FOS*, *JUN*, *SERPINE1*, and *PTGS2* were validated using an RT-qPCR approach. Their expression was consistent with the RNA-seq data, validating the reliability and accuracy of the sequencing approach (Appendix A).

## 4. Discussion

Our findings generally agree with our working hypotheses, showing that colder environmental temperatures influence physiological, morphological, and molecular responses in sheep. These effects were more pronounced in the cold-adapted Hulunbuir lambs compared to the Hu lambs, suggesting breed-specific adaptive responses to low temperatures. Below, we discuss how these changes appear at the phenotypic and molecular levels, highlighting physiological adaptations and enriched pathways associated with cold tolerance.

### 4.1. The Impact of Temperature on Physiological Traits, Wool and Skin Characteristics of the Lambs

The skin serves as a crucial barrier against external environmental factors and plays a key role in thermoregulation. In sheep, wool acts as an important appendage to the skin, providing insulation and protection against physical damage [1]. Our study revealed that exposure to a colder temperature induced changes in physiological traits, wool, and skin characteristics in both the Hulunbuir and Hu lambs, with more pronounced effects observed in the Hulunbuir lambs. Specifically, we observed that the rate of growth of the wool and epidermis thickness was greater in the Hulunbuir lambs compared to the Hu lambs at −20 °C. A previous study of yaks also revealed distinct gene expression patterns during different seasonal hair cycle phases that are associated with their adaptation to extreme climates [13], and this supports the idea that cold exposure induces changes in wool and skin characteristics in sheep. Similarly, Ames et al. [10] demonstrated that wool length influences the thermoneutral zone in sheep, with longer wool conferring greater cold tolerance, a finding that is consistent with our results.

Increased epidermis thickness can enhance the skin’s barrier function, providing protection against environmental stressors and improving overall thermal insulation. Previous research has indicated that skin thickness in breeds such as Scottish Blackface sheep and newborn Merino lambs is positively correlated with cold tolerance [1,29]. The long hair, thick skin, and fat layer of Qinghai-Tibetan sheep have been described as collectively forming an insulating layer that prevents heat loss and enhances cold tolerance [16]. It is therefore notable that, in our study, the epidermis thickness of the Hulunbuir lambs was greater at −20 °C compared to 15 °C, while the Hu lambs also exhibited increased epidermis thickness at −20 °C, but this difference was not significant.

It has been reported that Hulunbuir sheep have a higher proportion of kemp hair, with the mean fiber diameter of the kemp fiber as the greatest among all wool types [17,30]. In this study, both the Hulunbuir and Hu lambs exhibited an increase in the amount of kemp hair at −20 °C. The fiber diameters of both the coarse and fine wool in the Hulunbuir lambs increased slightly at −20 °C, although the increase in the Hu lambs was slightly lower than for the Hulunbuir lambs. Follicle density in the lambs at −20 °C was lower than at 15 °C, and, perhaps unsurprisingly, both the Hulunbuir and Hu lambs had an increase in wool yield in colder environments, with the wool yield of Hulunbuir lambs higher than that of the Hu lambs. Collectively, these adaptations may contribute to superior cold tolerance.

We also found that the rectal temperature and respiratory rate were decreased at −20 °C, suggesting that physiological adjustments were likely made by these lambs in response to the lower temperature. Li [31] found that the rectal temperature in 6-month-old Dupo and Suffolk lambs decreased after eight days of cold exposure, although the difference was not significant. A study of Sanhe cattle reported that the rectal temperature and respiratory rate were lower during winter months compared to spring and summer [32], which is consistent with our results. In contrast, a study of Altay and Hu lambs found that the rectal temperature at an environmental temperature of −5 °C was higher than at 20 °C, which may be attributed to the fact that these sheep did not experience any metabolic stress [33]. In this respect, the decrease in the rectal temperature and respiratory rate in colder environments may be adaptations to reduce energy demand to maintain temperature and decrease heat loss.

Overall, the combination of increased wool growth, increased epidermis thickness, higher wool yield, and reduced rectal temperature and respiratory rate may enhance the cold tolerance of Hulunbuir lambs, further highlighting their adaptations to colder environments. Although similar changes were observed in Hu lambs, which likely helped them accommodate the colder environment, the changes were not as pronounced as those observed for the Hulunbuir lambs.

### 4.2. Changes in the Immune and Endocrine System, Development and Regeneration, and Signal Transduction in Sheep in Colder Environments

The pathways associated with the colder environment across all comparison groups were enriched in the IL-17 signaling pathway that leads to the expression of pro-inflammatory cytokines, chemokines, and antimicrobial peptides, parathyroid hormone synthesis, secretion and action, the oxytocin signaling pathway, aldosterone synthesis and secretion, the TNF signaling pathway and signal transduction, and osteoclast differentiation likely involved in development and regeneration. Common DEGs in these pathways include *JUN*, *DUSP1*, *NR4A1*, *FOS*, *NR4A2*, *EGR1*, *PTGS2,* and *FOSB*, all of which were upregulated in the colder environment. The genes *JUN*, *FOS,* and *FOSB* encode transcription factors activated in various signaling pathways including those related to the immune and endocrine systems, development and regeneration, and signal transduction. The transcription factor *JUN* was demonstrated to play a role in the proliferation and differentiation of keratinocytes and in cytokine production. Mice lacking *JUN* expression in keratinocytes develop normal skin but have reduced EGFR levels in the eyelids, leading to them having open eyes at birth [34]. The expression of *JUN* is prominent in both hyperproliferating basal and suprabasal keratinocytes [35]. Our study found that the expression of *JUN* in the Hulunbuir and Hu lambs was greater at the lower temperature, and this suggests that the upregulation of *JUN* in colder environments may stimulate keratinocyte proliferation, potentially explaining the observed increase in epidermis thickness and enhanced cold tolerance.

The expression of *FOS* has been related to hair follicle development, particularly during the transition from birth to Er-mao in Chinese Tan sheep [36]. This aligns with our findings that suggest that *FOS* expression promotes wool development in both Hulunbuir and Hu lambs in colder conditions. The expression of *FOSB* has been associated with fibroblast proliferation and wound healing [37], suggesting a potential role in skin repair in response to cold-induced damage and stress. The upregulation of *FOSB* expression in colder environments may therefore assist in enhancing skin repair and cold adaptability. The NR4A receptors are a subfamily of nuclear receptors that function as ligand-independent transcription factors. They play a role in various cellular processes, including metabolism, cardiovascular and neurological functions, inflammation, and cancer. The NR4A subfamily includes NR4A1, NR4A2, and NR4A3 [38]. The NR4A receptors are induced by stimuli or stressors, and receptor NR4A1 plays a role in cellular homeostasis, synaptic remodeling, and metabolism [39]. Interestingly, in the context of cold adaptation, there is evidence for NR4A1 being a cold-induced effector of brown fat thermogenesis [40]. The zinc-finger transcription factor EGR1 is involved in various cellular processes like proliferation, differentiation, apoptosis, and inflammation. It is responsive to a wide range of signals, including mechanical, growth factors, and stress stimuli and can regulate transforming growth factor-β signaling and skin fibrosis, thus restoring tissue homeostasis [41]. EGR1 has been implicated in the proliferation of dermal papilla cells in Hu sheep [42]. Prostaglandin-endoperoxide synthase 2 (PTGS2) is an enzyme that catalyzes the production of prostaglandins that are involved in inflammation, pain, and fever. Zhao et al. [43] revealed that prostaglandins induce wool growth in Aohan fine wool sheep and that *PTGS2* is upregulated in wool-growing skin compared to wool-free skin. This supports the finding that *PTGS2* is associated with increased wool growth in the Hulunbuir and Hu lambs exposed to colder temperatures. The high expression of *DUSP1* can inhibit apoptosis and promote hair follicle stem cell proliferation [44]. The phosphatase enzyme encoded by *DUSP1* has also been shown to be involved in wool growth and its cycling [45].

In addition to the differences in gene expression observed between the −20 °C and 15 °C environments, the genes *FOSB*, *FOS*, *JUN*, *PTGS2,* and *DUSP1* showed greater fold-change differences in the Hulunbuir lambs, when compared to the Hu lambs. This suggests that these genes may play a greater role in enabling cold tolerance in the Hulunbuir lambs. In contrast, *NR4A1*, *NR4A2,* and *EGR1* exhibited greater fold-change differences in the Hu lambs, when compared to the Hulunbuir lambs, suggesting that their mechanism of action in cold adaptation may be different. This will require further investigation.

### 4.3. Special Adaptations of Signal Transduction, Membrane Transport, Excretory and Endocrine System, and Metabolism in Hulunbuir Sheep at −20 °C

In the colder environment at −20 °C, the unique DEGs of the Hulunbuir lambs were enriched in pathways related to signal transduction, including the apelin signaling pathway and the TNF signaling pathway, as well as osteoclast differentiation associated with development and regeneration. DEGs associated with ABC transporters, excretory systems, taurine and hypotaurine metabolism, and glycan biosynthesis and metabolism were also identified. It is notable that the TNF signaling pathway and osteoclast differentiation were present in the same pathways associated with the colder environment across all comparison groups, with these two pathways most enriched in the Hulunbuir lambs at −20 °C. This suggests that these pathways played a stronger role in helping the lambs cope with the colder environment.

Transcripts for the genes *SOCS3*, *IL1A*, *JUNB*, *CREB3L4*, *NOX1*, *FCGR2B*, *LIF*, and *TNFAIP3* were enriched. Each of these genes can be linked to responses to cold challenges. For example, *SOCS3* encodes a cytokine signal inhibitor that helps maintain immune homeostasis by blocking the signaling of various cytokines. It plays a role in the adaptation to colder environments after chick hatching [46]. The gene *IL1A* is expressed in a variety of cells, including macrophages, neutrophils, epithelial cells, endothelial cells, and fibroblasts, and in both healthy and inflammatory conditions. It has also been shown to be associated with the regulation of wool growth in Aohan fine wool sheep, with a speculated role in wool growth regulation during winter [47]. Transcription factor JunB plays a role in various physiological processes, including immune responses, cell proliferation, and tumorigenesis. Its dysregulation may impair hair follicle stem cells, indicating that it may be important in maintaining skin homeostasis [48]. The apelin signaling pathway was a key pathway identified in the Hulunbuir lambs in responding to the colder environment. This includes the genes *NOS3*, *KLF2*, *LIPE*, *RYR3*, *MYLK3,* and *SERPINE1*. *SERPINE1* is an early response-to-injury acute phase gene, the expression of which is upregulated by numerous growth factors, cytokines, and hormones, including tumor necrosis factor-α, transforming growth factor-β, interleukins, glucocorticoids, insulin, adrenaline, and angiotensin II [49]. *SERPINE1* not only plays a role in stress responses but is also associated with thrombosis, potentially affecting blood circulation in colder environments [50]. Research has suggested that *SERPINE1* promotes cashmere growth in Changthangi goats [51], and, similarly, the cold-adapted Yakutian horse has been found to exhibit the same adaptive mechanisms, with genes related to coat density, body size, lipid accumulation, and vascular contraction undergoing adaptive evolution [2].

## 5. Conclusions

This study identified differences in physiological traits, wool and skin characteristics, and transcriptional responses in Hulunbuir and Hu lambs when exposed to a temperature of −20 °C for 38 days. The Hulunbuir lambs exhibited more pronounced change and a greater number of DEGs, suggesting increased sensitivity to cold challenges. This breed appears to possess physiological and genetic adaptation capabilities that enhance its ability to withstand colder environments. These adaptations include increased wool growth, a thicker epidermis, decreased rectal temperatures and respiratory rates, and higher wool yields compared to the Hu lambs.

The TNF signaling pathway, which is related to signal transduction, and osteoclast differentiation, which is related to development and regeneration, were not only unique pathways activated with the cold challenge of the Hulunbuir lambs but were also part of the common enriched pathways. This suggests that these pathways may serve as key mechanisms for sheep to adapt to colder environments. We also found that unique pathways, including the apelin signaling pathway, were activated in the Hulunbuir lambs. Our findings provide a better understanding of the molecular mechanisms underlying cold adaptation in sheep, with potential implications for improving cold tolerance through genetic and management strategies.

## Figures and Tables

**Figure 1 animals-15-01405-f001:**
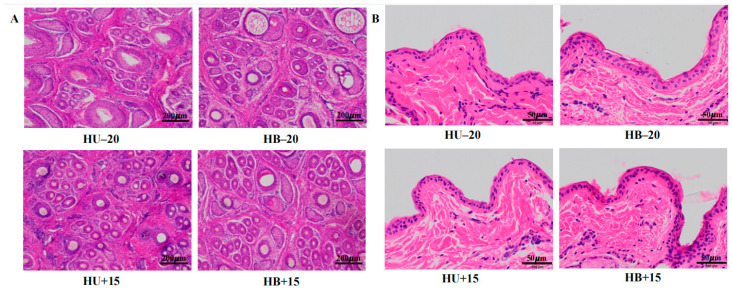
Staining (hematoxylin and eosin) of skin tissue from Hulunbuir and Hu lambs at −20 °C and 15 °C: (**A**) Hair follicle density at −20 °C and 15 °C. (**B**) Epidermis thickness at −20 °C and 15 °C.

**Figure 2 animals-15-01405-f002:**
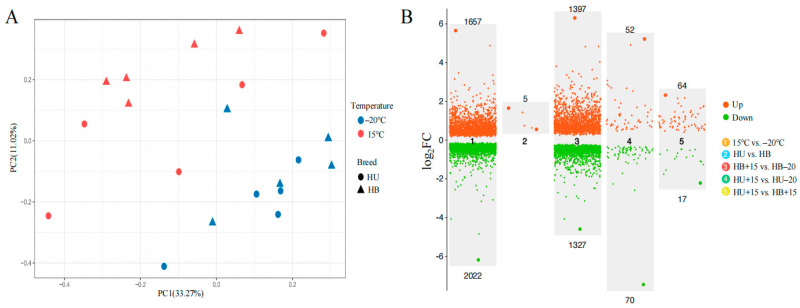
Comparative analyses of genetic architecture and differential gene expression between Hulunbuir lambs and Hu lambs at −20 °C and 15 °C: (**A**) A principal component analysis of the Hu (HU) and Hulunbuir (HB) lambs at −20 °C and 15 °C. The percentages on the axes represent the proportion of total variance in the original data explained by each principal component. (**B**) A volcano plot of the number of DEGs in five comparisons. The upper and lower numbers represent upregulated and downregulated DEGs, respectively, in the five comparisons shown. HB+15: Hulunbuir lambs at 15 °C; HB−20: Hulunbuir lambs at −20 °C; HU+15: Hu lambs at 15 °C; and HU−20: Hu lambs at −20 °C.

**Figure 3 animals-15-01405-f003:**
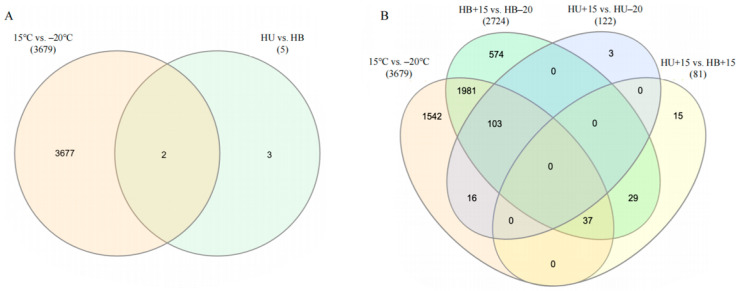
Venn diagram of differentially expressed genes in comparisons of the lamb groups: (**A**) A breed versus temperature group comparison. (**B**) Cross-comparisons of four groups (15 °C vs. −20 °C, HB+15 vs. HB−20, HU+15 vs. HU−20, and HU+15 vs. HB+15). Numbers beneath each comparison represent the total number of DEGs. HU: Hu lambs; HB: Hulunbuir lambs; HB+15: Hulunbuir lambs at 15 °C; HB−20: Hulunbuir lambs at −20 °C; HU+15: Hu lambs at 15 °C; and HU−20: Hu lambs at −20 °C.

**Figure 4 animals-15-01405-f004:**
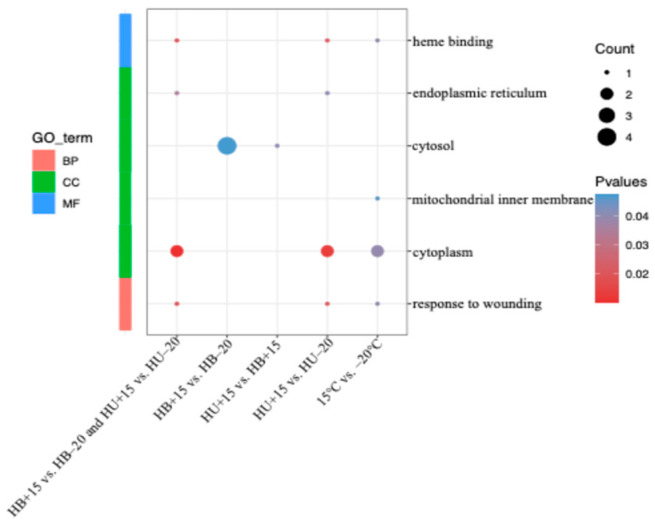
GO analysis results of the DEGs in various comparisons. BP: biological process; CC: cellular component; and MF: molecular function. The significance level is represented by the color of each point with redder hues indicating greater significance. HB+15: Hulunbuir lambs at 15 °C; HB−20: Hulunbuir lambs at −20 °C; HU+15: Hu lambs at 15 °C; and HU−20: Hu lambs at −20 °C.

**Figure 5 animals-15-01405-f005:**
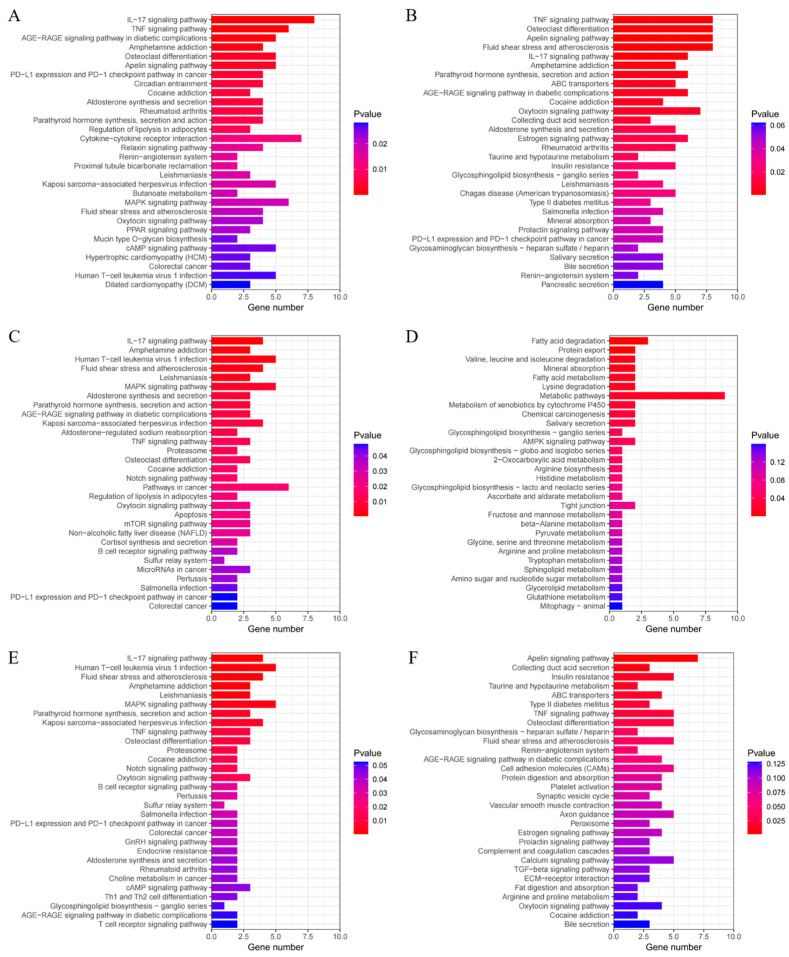
Top 30 KEGG analysis results of the DEGs in various group comparisons: (**A**) KEGG analysis results of the DEGs in the temperature comparison groups; (**B**) KEGG analysis results of the DEGs in the HB+15 vs. HB−20 group; (**C**) KEGG analysis results of the DEGs in the HU+15 vs. HU−20 group; (**D**) KEGG analysis results of the DEGs in the HU+15 vs. HB+15 group; (**E**) KEGG analysis results of the common DEGs in the HB+15 vs. HB−20 and HU+15 vs. HB−20 groups; (**F**) KEGG analysis results of the unique DEGs in the HB+15 vs. HB−20 group.

**Figure 6 animals-15-01405-f006:**
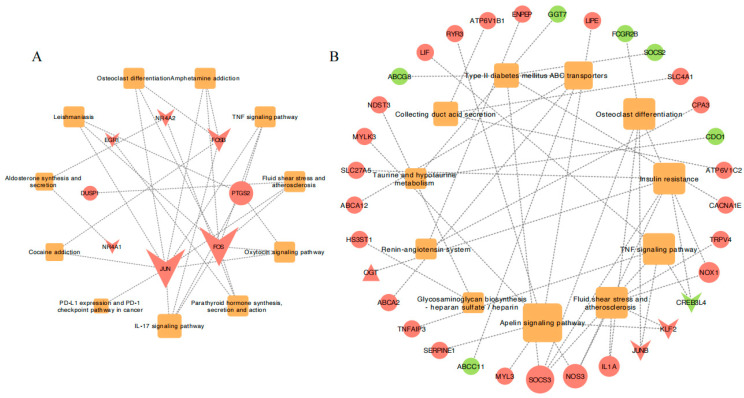
The gene co-expression networks for various comparisons: (**A**) Common gene co-expression networks in 11 pathways associated with the cold challenge across all comparison groups. (**B**) Unique gene co-expression networks in the HB+15 vs. HB−20 comparison. The color of the nodes represents genes (upregulation in red, downregulation in green) or pathways (orange). The shapes represent genes (circles), transcription factors (inverted triangles), transcription cofactors (triangles), or pathways (squares).

**Figure 7 animals-15-01405-f007:**
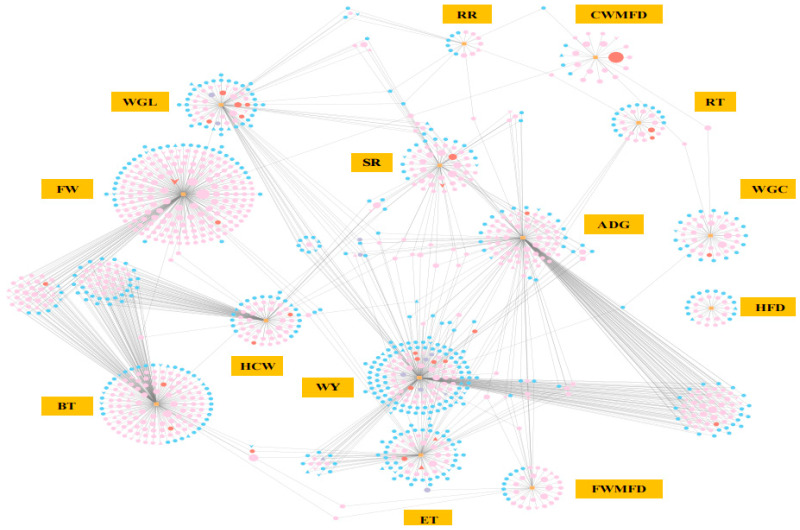
Phenotype and gene co-expression networks at −20 °C. The color of the nodes represents the 2555 unique DEGs in a comparison of HB+15 vs. HB−20 (pink), common 103 DEGs in a comparison of HB+15 vs. HB−20 and HU+15 vs. HU−20 (red), the 19 unique DEGs in HU+15 vs. HU−20 (purple), other DEGs (blue), and phenotypes (orange). The shapes represent genes (circles), transcription factors (inverted triangles), cofactors (triangles), or phenotypes (rectangles). WGL: wool growth length (cm); CWMFD: coarse wool mean fiber diameter (µm); FWMFD: fine wool mean fiber diameter (µm); FW: the proportion of fine wool fibers (%); WY: wool yield (g); WGC: wool grease content (%); HFD: hair follicle density (per mm^2^); ET: epidermis thickness (µm); ADG: average daily gain (g/d); BT: backfat thickness (mm); HCW: hot carcass weight (kg); SR: slaughter rate (%); RT: rectal temperature (°C); and RR: respiratory rate (breaths/min).

**Table 1 animals-15-01405-t001:** Diet composition and nutrient level.

Ingredient		Nutrient ^2^	
Alfalfa hay (%)	25.35	DM (%)	96.26
Rice husk (%)	4.65	ME (MJ/kg)	10.28
Corn (%)	31.70	Crude protein (%)	14.01
Cottonseed meal (%)	2.30	Ca (%)	0.54
Soybean meal (%)	12.50	P (%)	0.26
Corn starch (%)	18.50		
Bypass fat powder (%)	3.50		
Limestone (%)	0.30		
NaCl (%)	0.70		
Premix ^1^	0.50		
Rate of concentrate to roughage	70:30		

^1^ The main components of the premix are vitamins and trace elements (Cu, Fe, Mn, Zn, I, Se). ^2^ DM: dry matter; ME: metabolizable energy. ME is a calculated value; all other nutrient levels are measured.

**Table 2 animals-15-01405-t002:** Primers used for RT-qPCR analysis.

Gene Name	Primer Sequence (5′-3′)	Product Size (bp)	Tm (°C)
*ACTB*	GGTCATCACCATCGGCAAT	113	60
	CGTAGAGGTCTTTGCGGATG		
*FOS*	TACAATGGCTAGTGCAGCCC	163	60
	TTGGTCTGTCTCCGCTTGGA		
*JUN*	GCTTCCAAGTGCCGGAAAAG	184	60
	GCTGCGTTAGCATGAGTTGG		
*PTGS2*	TAACACGCTCTACCACTGGC	176	60
	GATTCCTACGACCAGCGACC		
*SERPINE1*	AAGAGCACCGTCCAGAGAGA	130	60
	ACATCTGCATCCTGAATTTCTCAA		

**Table 3 animals-15-01405-t003:** ANOVA-derived differences in various characteristics of Hulunbuir and Hu lambs at different temperatures.

Characteristic	15 °C	−20 °C	Hu	Hulunbuir	SEM	*p*-Value		
						Temperature	Breed	Interaction
WGL (cm)	1.1 ± 0.04	1.7 ± 0.10	1.3 ± 0.09	1.4 ± 0.15	0.09	<0.001	0.243	0.046
FWMFD (µm)	20.5 ± 0.58	21.1 ± 0.42	21.7 ± 0.40	19.8 ± 0.41	0.35	0.344	0.004	0.262
WY (g)	0.3 ± 0.02	0.3 ± 0.03	0.3 ± 0.02	0.3 ± 0.03	0.02	0.140	0.037	0.139
HFD (/mm^2^)	38.7 ± 2.39	33.0 ± 1.45	33.5 ± 1.74	38.1 ± 2.31	1.51	0.044	0.094	0.208
ET (µm)	15.2 ± 0.65	18.7 ± 0.79	16.5 ± 0.60	17.3 ± 1.15	0.64	0.002	0.431	0.050
ADG (g/d)	105.5 ± 17.03	59.6 ± 10.44	75.2 ± 17.44	89.8 ± 14.14	11.05	0.037	0.480	0.286
RT (°C)	38.8 ± 0.04	38.1 ± 0.09	38.4 ± 0.14	38.4 ± 0.12	0.09	<0.001	0.923	0.923
RR (breaths/min)	22.4 ± 1.42	16.0 ± 0.84	18.7 ± 1.36	19.7 ± 1.75	1.09	0.001	0.526	0.213

WGL: wool growth length; FWMFD: fine wool mean fiber diameter; WY: wool yield; HFD: hair follicle density; ET: epidermis thickness; ADG: average daily gain; RT: rectal temperature; and RR: respiratory rate. *p*-values derived from the two-way ANOVA models. The interaction refers to the combined effect of temperature and breed on the variable.

**Table 4 animals-15-01405-t004:** LSD post hoc comparison-derived differences in various characteristics of the groups of Hulunbuir and Hu lambs at the two temperatures.

Characteristic	−20 °C		15 °C		*p*-Value			
	HU−20	HB−20	HU+15	HB+15	HU+15 vs. HU−20	HB+15 vs. HB−20	HU−20 vs. HB−20	HU+15 vs. HB+15
WGL (cm)	1.5 ± 0.12	1.8 ± 0.13	1.1 ± 0.08	1.0 ± 0.46	0.028	<0.001	0.096	0.310
FWMFD (µm)	21.7 ± 0.69	20.4 ± 0.35	21.8 ± 0.48	19.2 ± 0.67	0.903	0.147	0.144	0.014
WY (g)	0.3 ± 0.15	0.4 ± 0.03	0.3 ± 0.03	0.3 ± 0.03	0.999	0.074	0.025	0.630
HFD (/mm^2^)	32.4 ± 2.20	33.6 ± 2.11	34.7 ± 2.87	42.7 ± 3.06	0.549	0.041	0.702	0.092
ET (µm)	17.3 ± 0.64	20.0 ± 1.20	15.8 ± 0.97	14.6 ± 0.87	0.240	0.006	0.076	0.370
ADG (g/d)	41.2 ± 7.16	77.9 ± 16.44	109.3 ± 27.16	101.6 ± 23.66	0.042	0.435	0.074	0.837
HCW (kg)	19.5 ± 0.47	19.8 ± 0.72	21.1 ± 0.39	20.2 ± 0.62	0.030	0.732	0.736	0.221
RT (°C)	38.1 ± 0.16	38.1 ± 0.10	38.8 ± 0.07	38.8 ± 0.04	0.004	<0.001	0.918	1.000
RR (breaths/min)	16.5 ± 1.37	15.5 ± 1.08	20.8 ± 2.05	24.0 ± 1.89	0.122	0.004	0.559	0.284

WGL: wool growth length; FWMFD: fine wool mean fiber diameter; WY: wool yield; HFD: hair follicle density; ET: epidermis thickness; ADG: average daily gain; HCW: hot carcass weight; RT: rectal temperature; and RR: respiratory rate. *p*-values derived from LSD post hoc tests.

## Data Availability

Data are made available upon reasonable request.

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
