# Peer review of "Transcriptomic Analysis of Skin Tissue Reveals Molecular Mechanisms of Thermal Adaptation in Cold-Exposed Lambs"

_animals, 2025, doi:10.3390/ani15101405_

Round 1

Reviewer 1 Report

Comments and Suggestions for Authors

Manuscript can be interesting, however almost half of the results are given or shown in supplementary data (Which I can access) therefore, I can’t review this manuscript. Based on the limited results presented inside the manuscript, here are my concerns about the manuscript;

  1. Simple summary can be at large based on Table 3 and 4 data.
  2. The description and morphometric data of both breeds are missing.
  3. I also suggest inclusion of the pictures of both breeds, their population distribution in respective geographic area.
  4. I can’t see the supplementary data here at the peer review platform. Authors have skin tissues, presenting Hematoxylin and Eosin (H&E) Staining or Masson’s Trichrome Staining would visualize the phenotype on which this article is based. If these images are present in supplementary data, shift it to the main manuscript.
  5. Rate of concentrate to roughage of 70 : 30 at both temperature experiments signifies a limitation of experimental design, which shall be justified in Abstract and Discussions & conclusions.
  6. Authors can use a higher FC cutoff (e.g., ≥ 2) for downstream analysis (e.g., pathway enrichment), in order to narrow down the pathways and its respective networks of interests.
  7. It is must to provide all KEGG pathway analysis figures in the main body of the manuscript.
  8. Manuscript and especially the results description can be improved i.e., Line 47: “. Similar changes 47 were observed for the Hu lambs.” please elaborate.
  9. Spelling and grammar mistakes shall be addressed i.e., in table 1 “consentrate”.

Author Response

Comment 1: Simple summary can be at large based on Table 3 and 4 data.

AU: We have now made Tables 3 and 4 into supplementary tables, using the text to describe the significant results from the ANOVAs and the post-hoc tests.

Comment 2: The description and morphometric data of both breeds are missing.

AU: We have added descriptions of the Hulunbuir and Hu breeds. This has lengthened the introduction, with this being offset by text deletion elsewhere.

Comment 3: I also suggest inclusion of the pictures of both breeds, their population distribution in respective geographic area.

AU: We are wary about including pictures of the lambs in the manuscript text as it invites unnecessary comparison that is subjective, but here follows images of individual experimental lambs from the two breeds. We have however added detailed descriptions of the breeds in the introduction section.

Comment 4: I can’t see the supplementary data here at the peer review platform. 

AU: That is beyond our control unfortunately, but we will remind the journal of the need to make these results available to reviewers.

Comment 5: Authors have skin tissues, presenting Hematoxylin and Eosin (H&E) Staining or Masson’s Trichrome Staining would visualize the phenotype on which this article is based. If these images are present in supplementary data, shift it to the main manuscript.

AU: We have added these results to the main body of the text now.

Comment 6: Rate of concentrate to roughage of 70 : 30 at both temperature experiments signifies a limitation of experimental design, which shall be justified in Abstract and Discussions & conclusions.

AU: The 70% roughage to 30% concentrate ratio in the lamb diets was designed to achieve the following objectives, aligning with both nutritional requirements and experimental goals:

Metabolizable Energy (ME) and Protein Balance: Roughage provides structural carbohydrates critical for rumen health and microbial fermentation, while concentrates deliver high-energy density and digestible protein. The 30% concentrate ensured sufficient metabolizable energy intake (11.0 MJ/d) and crude protein (164.2 g/d) to meet the growth demands of lambs under cold stress, supporting a target daily weight gain of 100 g as per the Chinese Meat Sheep Feeding Standard (NY/T 816-2004).

Fiber Requirement for Rumen Function: A 70% roughage ratio maintained adequate neutral detergent fiber (NDF) and acid detergent fiber (ADF) levels to sustain rumen motility, pH stability, and microbial activity.  This is essential for preventing metabolic disorders (e.g., acidosis) and optimizing volatile fatty acid (VFA) production, particularly propionate and butyrate, which are critical for energy metabolism. A fixed ratio minimized variability in nutrient intake between temperature groups (-20°C vs. 15°C), ensuring that observed physiological differences could be attributed primarily to cold stress rather than dietary fluctuations.

Compliance with Feeding Standards: The ratio aligns with NY/T 816-2004, which recommends balanced roughage-to-concentrate ratios for fattening lambs to achieve optimal growth performance and carcass quality under controlled conditions.

The text has been adjusted to reflect this.

Comment 7: Authors can use a higher FC cutoff (e.g., ≥ 2) for downstream analysis (e.g., pathway enrichment), in order to narrow down the pathways and its respective networks of interests.

AU: We could, but we think it is “early days” for this kind of work and hence the lower cutoff to enable more pathways to be found.

Comment 8: It is must to provide all KEGG pathway analysis figures in the main body of the manuscript.

AU: We have added these to the main body of the manuscript text as figures 4 and 5.

Comment 9: Manuscript and especially the results description can be improved i.e., Line 47: “. Similar changes 47 were observed for the Hu lambs.” please elaborate.

AU: We have revised this wording.

Comment 10: Spelling and grammar mistakes shall be addressed i.e., in table 1 “consentrate”.

AU: The spelling and grammar have been revised, and all changes to the text have been marked in red text colour.

Reviewer 2 Report

Comments and Suggestions for Authors

Dear authors

Mengyu Feng et al in this work have analyzed molecular mechanisms of thermal adaptation in cold-exposed lambs. Hulunbuir and Hu ewe lambs were feeding at -20°C and 15°C. The groups were maintained at these temperatures for 38 days. Skin tissues were analyzed with transcriptome sequencing, and selected wool and physiological traits were assessed. The Hulunbuir lambs at -20°C had greater wool length growth and epidermis thickness, but lower hair follicle density, rectal temperature and respiratory rate, when compared to the Hulunbuir lambs at 15°C. Similar changes were observed for the Hu lambs. Transcriptome analyses revealed the activation of pathways related to immune and endocrine systems, signal transduction, and development and regeneration in both breeds at -20°C. The TNF signaling pathway and osteoclast differentiation, may play roles in cold-adaptation, as they are associated with differentially expressed genes identified in the Hulunbuir lambs, as well as shared DEGs between both breeds. This study revealed physiological and molecular differences in lambs exposed to lower temperatures, while also suggesting potential targets for improving cold-tolerance, welfare and productivity. The data is encouraging and certainly suitable for publication. However, a number of queries have arisen during my review, which I list below for the authors to consider.

  1. The rectal temperature of sheep shows a distinct circadian rhythm. The author should clearly define the time for measuring rectal temperature.
  2. Although the gastrointestinal space of sheep is limited, cold stress does increase feed intake, which directly affect the nutritional supply of sheep, thereby influencing their adaptability in low-temperature environments. The author provided the daily feeding strategy and feed supply quantity, and should provide the daily feed intake of sheep.

Author Response

Comment 1: The rectal temperature of sheep shows a distinct circadian rhythm. The author should clearly define the time for measuring rectal temperature.

AU: The sentence: Rectal temperature measurements were conducted at 7:00 AM (prior to feeding) to standardize the timing and minimize confounding effects of circadian rhythm or postprandial metabolic changes, has been added.

Comment 2: Although the gastrointestinal space of sheep is limited, cold stress does increase feed intake, which directly affect the nutritional supply of sheep, thereby influencing their adaptability in low-temperature environments. The author provided the daily feeding strategy and feed supply quantity, and should provide the daily feed intake of sheep.

AU: See comments addressing concerns of reviewer #1 and in the manuscript where we have discussed issue.

Reviewer 3 Report

Comments and Suggestions for Authors

ANIMALS - MDPI

Referee’s Evaluation Report

MANUSCRIPT IDENTIFICATION: animals-3606668

Transcriptomic analysis of skin tissue reveals molecular mechanisms of thermal adaptation in cold-exposed lambs 

 (ORIGINAL ARTICLE)

Comments to Authors/Editor:

The paper of Feng & colleagues aimed to investigate the molecular mechanisms of wool and skin responses to cold stress in lambs belonging to two breeds with varying cold tolerance. This interesting transcriptomic study falls within the scope of ANIMALS, while it is sufficiently informative for the replication of the study. In general, the organization of the experiment seems to be well designed, yet the sentence structure and grammar must be improved; the English structure is certainly fragile. In the title, L33, “… This finding suggests their… or “These findings suggest their…????; correct accordingly along with the entire manuscript. In the Abstract section, and along with the whole manuscript, I suggest to change the abbreviations of the treatments: Hulunbuir at -20°C and at 15°C as “HB-20” & “HB15” as well as Hu at -20°C and at 15°C as “HU-20” & “HU15”, respectively; please rewrite along with the whole manuscript, tables and figures. Please include the age and live weights (i.e., initial & final) of lambs as well as location in the Abstract. L43 please change from “The Hulunbuir lambs at -20°C”.., to “The HB-20 lambs…”; correct accordingly along with the whole manuscript. L50, change from “… and regeneration in both breeds at -20°C.., to …and regeneration at -20°C, irrespectively of breed. L54, change from ”…while also suggesting…”, to  “… while also suggests…”. The Introduction section is generally long, and its structure and content are certainly not the best option to introduce the research question to the readership. While any statement must include a citation (i.e., L59-63), the authors must contextualize not only the importance of Asia regarding the world sheep population, but also another key issues from a social, economic, and productive point of view, especially with respect to wool/fiber-producing breeds used in the study. The Introduction section is quite large; some information must be moved in the Discussion (i.e., L69-98). While the authors defined the aims of the study, they do not proposed the working hypothesis in the Introduction section; this is a must. In the Material & Methods section, the authors started this section with the institutional approval of the study, yet the authors must state if the study followed any international guide for the use of animals in research; this is a must. In general, the M&M section was written in a carless fashion. The Section 2.1., must include “Location, animals and their management”. Please include in this part the location where the study was carried out including information regarding the latitude, longitude, altitude, prevalent photoperiod and other environmental conditions. Then, the authors should try:  Lambs (n=36) obtained from the Darhan-Muminggan Joint County Farm in inner Mongolia, of similar age (i.e., 8 mo. old) and belonging to the breeds Hulumbuir  (n=10; 34.5 ± 0.70 kg) and  Hu (n=10;  34.9 ± 0.79 kg) were raised at the Ecological and Agricultural Experimental Station, Gaolan, Gansu Province, People´s Republic of China. The lambs of each breed were….”.  In this 2.1. section, the authors must briefly include information regarding the nutritional and reproductive (if any) management of the lambs; this is central to better understand the main research outcomes of this study. The section 2.2. mut include “Experimental Design and Treatments. Then, continue with the section 2.3. “Sample collection and physiological response variables”. Then, subheading 2.4. Wool and skin response variables”. 2.5. “Transcriptome sequencing and differentially expressed gene analyses”. 2.6. “Functional enrichment and co-expression module analyses”. 2.7. “Validation with reverse transcription-quantitative PCR (RT-qPCR)”., and finally, 2.8. “Statistical analyses”.

In the Results Section, the authors include the same information in the text and in the tables; it is really tiring and boring. Moreover, it makes no sense to include all the variables studied according to each “source of variation” (i.e., factors; breed or temperature) especially if no variation among factors occurred. My recommendation is, if there are no differences between in-class factors (i.e., temperature and breed), the authors should only indicate in the text the average of the factor within the class, remove the "unaffected" variables in the tables, and include the factor values within class and between classes in the supplementary information. Thus, in the Results section, the tables would only include the variables in which there was a statistical difference between factors, leaving one table for simple effects and another table for interaction purposes, but only considering the variables that were significantly affected (i.e., p<0.05). In this way, a "visual pollution" would be avoided since the tables would only include the variables that did differ statistically by the simple effect or by the interaction; for the other variables only include in the text the general average for each response variable by factor (i.e., breed or temperature).  Besides, the same is true for table 4; just include the response variables that statistically differed in-class factors. Also, avoid including information already presented in the M&M section. In the Results section the authors must include just the significant results obtained, just that, nothing else; correct accordingly. Further, avoid the use of the word “Phenotypes” in the tables, use Variables, instead. The quality assessment of RNM-Seq data and genome alignment, Gene expression level and cluster analyses, GO analysis of the DEGs, KEGG analysis of DEGs are clear. Please, consider the previous comments and never forget that in science and politics, “form” is “substance”. Therefore, the way (i.e., “form”) in which the results are presented speaks for itself of the scientific quality of any research group. In general, the novelty value of the results is reasonable. In this section the authors presented their main research outcomes in diverse Figures and Tables. As mentioned, the authors must avoid the use of both the word “significant” and the “probability value itself”; it is a pleonasm. Regarding to the Discussion section, I think the way the authors used to initiate this section, was not the best strategy to do. At the beginning of the Discussion, I do strongly suggest initiating this section including the working hypothesis of the study. The authors must define if, with the obtained results, such hypothesis is rejected or non-rejected. For this reason, the authors must include the working hypothesis prior to the objectives in the Introduction section. After this opening paragraph, the authors must follow the same order in this section according to that proposed in the Results section. The authors must link, in a logical fashion, their main findings along with the discussion section, to compare & to discuss and, thereafter, be able to propose some possible explanations for such specific outcomes, considering to previous similar studies from the scientific literature. In general, the authors made an accurate interpretation of the main findings. The authors must focus their main findings and confront them with respect to the scientific literature in a logical and focused fashion.  In general, the main outcomes of the study were well presented.  Preferably, the authors must to evade the use of paragraphs with such a discordant size or number of lines per paragraph (i.e., L520-531 vs. L559-561) and (i.e., L591-L594 vs. L595-L606). The Conclusions are fair; the use of the word “phenotypes” not only in the Conclusion section but along with the whole manuscript is not fortunate. The list of references cited in the manuscript is proper. This is a very interesting study. Yet, the authors must improve the clarity and logical arrangement of the main outcomes, especially in the Introduction, Results and Discussion sections.  The authors must align the conclusions regarding to the working hypothesis as well as the scientific question they try to solve; nothing else, just that. All the commented issues and requests should be clearly addressed by the authors. At this point, and based on the above comments, my pronouncement is that this manuscript cannot be accepted in its actual format; it requires major adjustments.

Author Response

Comments and Suggestions for Authors

The paper of Feng & colleagues aimed to investigate the molecular mechanisms of wool and skin responses to cold stress in lambs belonging to two breeds with varying cold tolerance. This interesting transcriptomic study falls within the scope of ANIMALS, while it is sufficiently informative for the replication of the study. In general, the organization of the experiment seems to be well designed, yet the sentence structure and grammar must be improved; the English structure is certainly fragile.

AU: The use of English has been checked and revised throughout the manuscript.

In the title, L33, “… This finding suggests their… or “These findings suggest their…????; correct accordingly along with the entire manuscript.

AU: Wording corrected

In the Abstract section, and along with the whole manuscript, I suggest to change the abbreviations of the treatments: Hulunbuir at -20°C and at 15°C as “HB-20” & “HB15” as well as Hu at -20°C and at 15°C as “HU-20” & “HU15”, respectively; please rewrite along with the whole manuscript, tables and figures.

AU: This is a great suggestion, albeit we have chosen to change to HB+15, HB-20, HU+15 and HU-20 to even better reflect the two breeds and temperatures in the experiment. We trust this is okay.

Please include the age and live weights (i.e., initial & final) of lambs as well as location in the Abstract.

AU: Done

L43 please change from “The Hulunbuir lambs at -20°C”.., to “The HB-20 lambs…”; correct accordingly along with the whole manuscript.

AU: As much as is practically possible, and as described in the point made above, we have followed this convention.

L50, change from “… and regeneration in both breeds at -20°C.., to …and regeneration at -20°C, irrespectively of breed.

AU: Wording changed

L54, change from ”…while also suggesting…”, to  “… while also suggests…”.

AU: Wording changed

The Introduction section is generally long, and its structure and content are certainly not the best option to introduce the research question to the readership. While any statement must include a citation (i.e., L59-63), the authors must contextualize not only the importance of Asia regarding the world sheep population, but also another key issues from a social, economic, and productive point of view, especially with respect to wool/fiber-producing breeds used in the study.

AU: We have extensively re-written and shortened the text in the introduction.

The Introduction section is quite large; some information must be moved in the Discussion (i.e., L69-98).

AU: See above. Section extensively re-written and shortened.

While the authors defined the aims of the study, they do not proposed the working hypothesis in the Introduction section; this is a must.

 AU: We have re-written and strengthened the wording explaining the main hypotheses.

In the Material & Methods section, the authors started this section with the institutional approval of the study, yet the authors must state if the study followed any international guide for the use of animals in research; this is a must.

AU: We have revised our ethics approval statement to say that: The Animal Welfare and Experimental Ethics Committee of the Northwest Institute of Ecological Environment and Resources, Chinese Academy of Sciences, approved all experimental procedures (CAS201810082) with these complying with People’s Republic of China GB/T 35892-2018 Laboratory Animal Welfare standards, and meeting ARRIVE (Animal Research: Reporting of In Vivo Experiments) 2.0 guidelines.

In general, the M&M section was written in a carless fashion. The Section 2.1., must include “Location, animals and their management”. Please include in this part the location where the study was carried out including information regarding the latitude, longitude, altitude, prevalent photoperiod and other environmental conditions.

Then, the authors should try:  Lambs (n=36) obtained from the Darhan-Muminggan Joint County Farm in inner Mongolia, of similar age (i.e., 8 mo. old) and belonging to the breeds Hulumbuir  (n=10; 34.5 ± 0.70 kg) and  Hu (n=10;  34.9 ± 0.79 kg) were raised at the Ecological and Agricultural Experimental Station, Gaolan, Gansu Province, People´s Republic of China. The lambs of each breed were….”.  In this 2.1. section, the authors must briefly include information regarding the nutritional and reproductive (if any) management of the lambs; this is central to better understand the main research outcomes of this study.

AU: Changes made

The section 2.2. mut include “Experimental Design and Treatments.

AU: Changed as suggested, and see comments addressing concern of reviewer #1.

Then, continue with the section 2.3. “Sample collection and physiological response variables”. 

AU: Changed as suggested

Then, subheading 2.4. Wool and skin response variables”. 2.5. “Transcriptome sequencing and differentially expressed gene analyses”. 2.6. “Functional enrichment and co-expression module analyses”. 2.7. “Validation with reverse transcription-quantitative PCR (RT-qPCR)”., and finally, 2.8. “Statistical analyses”.

AU: Changed as suggested

 In the Results Section, the authors include the same information in the text and in the tables; it is really tiring and boring. Moreover, it makes no sense to include all the variables studied according to each “source of variation” (i.e., factors; breed or temperature) especially if no variation among factors occurred. My recommendation is, if there are no differences between in-class factors (i.e., temperature and breed), the authors should only indicate in the text the average of the factor within the class, remove the "unaffected" variables in the tables, and include the factor values within class and between classes in the supplementary information.

Thus, in the Results section, the tables would only include the variables in which there was a statistical difference between factors, leaving one table for simple effects and another table for interaction purposes, but only considering the variables that were significantly affected (i.e., p<0.05). In this way, a "visual pollution" would be avoided since the tables would only include the variables that did differ statistically by the simple effect or by the interaction; for the other variables only include in the text the general average for each response variable by factor (i.e., breed or temperature).

AU: Great suggestion. We have accommodated this by making table 3 and 4 into supplementary tables (which can be accessed if readers required greater detail) and by describing in only the text the statistically significant differences in the results.

Besides, the same is true for table 4; just include the response variables that statistically differed in-class factors.

AU: See above comment.

Also, avoid including information already presented in the M&M section.

AU: The results have been extensively re-written and presented differently

In the Results section the authors must include just the significant results obtained, just that, nothing else; correct accordingly.

AU: Corrected – see revised text.

Further, avoid the use of the word “Phenotypes” in the tables, use Variables, instead.

AU: We have deleted the use of phenotype and replaced it with either ‘variable’ or ‘characteristic’ in the manuscript

The quality assessment of RNM-Seq data and genome alignment, Gene expression level and cluster analyses, GO analysis of the DEGs, KEGG analysis of DEGs are clear. Please, consider the previous comments and never forget that in science and politics, “form” is “substance”. Therefore, the way (i.e., “form”) in which the results are presented speaks for itself of the scientific quality of any research group. In general, the novelty value of the results is reasonable. In this section the authors presented their main research outcomes in diverse Figures and Tables. As mentioned, the authors must avoid the use of both the word “significant” and the “probability value itself”; it is a pleonasm.

AU: Agreed – but subject to the requirements of Reviewer #1 we have now put the KEGG results in the main body of the manuscript text.

Regarding to the Discussion section, I think the way the authors used to initiate this section, was not the best strategy to do. At the beginning of the Discussion, I do strongly suggest initiating this section including the working hypothesis of the study.

AU: We have added research hypotheses and a discussion statement to this effect.

The authors must define if, with the obtained results, such hypothesis is rejected or non-rejected. For this reason, the authors must include the working hypothesis prior to the objectives in the Introduction section.

AU: These have been added to the introduction.

After this opening paragraph, the authors must follow the same order in this section according to that proposed in the Results section. The authors must link, in a logical fashion, their main findings along with the discussion section, to compare & to discuss and, thereafter, be able to propose some possible explanations for such specific outcomes, considering to previous similar studies from the scientific literature.

AU: We have re-written and modified the discussion in many places to reflect the order of the results.

In general, the authors made an accurate interpretation of the main findings. The authors must focus their main findings and confront them with respect to the scientific literature in a logical and focused fashion.  In general, the main outcomes of the study were well presented. 

AU: Thank you.

Preferably, the authors must to evade the use of paragraphs with such a discordant size or number of lines per paragraph (i.e., L520-531 vs. L559-561) and (i.e., L591-L594 vs. L595-L606).

AU: We have condensed some of the smaller paragraphs into larger paragraphs of similar theme.

The Conclusions are fair; the use of the word “phenotypes” not only in the Conclusion section but along with the whole manuscript is not fortunate.

AU: This wording has been revised as described above.

The list of references cited in the manuscript is proper.

This is a very interesting study. Yet, the authors must improve the clarity and logical arrangement of the main outcomes, especially in the Introduction, Results and Discussion sections.  The authors must align the conclusions regarding to the working hypothesis as well as the scientific question they try to solve; nothing else, just that. All the commented issues and requests should be clearly addressed by the authors. At this point, and based on the above comments, my pronouncement is that this manuscript cannot be accepted in its actual format; it requires major adjustments. 

AU: We feel we have addressed this summary of the issues to be resolved in our comments above.

Round 2

Reviewer 1 Report

Comments and Suggestions for Authors

Authors have addressed all my comments.

Author Response

Authors have addressed all my comments.

AU: Thank you

Reviewer 3 Report

Comments and Suggestions for Authors

ANIMALS MANUSCRIPT ID: 3606668 – R1;  REVIEWER 2

Transcriptomic analysis of skin tissue reveals molecular mechanisms of thermal adaptation in cold-exposed lambs

REVIEWER: This is the R1 (i.e., Revised 1 version) of the paper of Feng & colleagues; they aimed to investigate the molecular mechanisms of wool and skin responses to cold stress in lambs belonging to two breeds with varying cold tolerance. In general, the organization of the experiment seems to be well designed, while the sentence structure and grammar were certainly improved; nonetheless, some key suggestion previousy proposed for this Reviewer were not addressed as required. The Introduction section is still quite long and the authors instead of contextualize their study including the importance of Asia and China regarding the world sheep population and another key issues from a social, economic, and productive point of view, especially with respect to wool/fiber-producing breeds used in the study, they included, instead, phenotipic information of both breeds which does not add any scientific power to the paper. The authors must contextualize the importance of they research with a global perspective. This Introduction section must be shortened; this is a must. Most of the suggestions proposed to enhance the sections Material and Methods as weel as Results were well approached, but there ae some issues that must be addressed. L288-290; please rewrite, this is a non-sense sentece structure. Again, the authors do not define when there were simple effects or interaction effects. If there is NO interaction effect, meaning that a factor A does not depend on a factor B, then they will need to address the simple effects for the affected response variables. If, on the contrary, the ANOVA defines an interaction effect, it means that factor A depends on factor B. Specifically, generate a subheading for each scenario; one subeading for single effects and another subheading for interaction efects, and include the respective Tables including only the significant effects in the Results section, and the non-significant ones may be included in a different Table(s) in the supplementary information.  This had already been required, but the authors did not solve it in the best way. The quality assessment of RNM-Seq data and genome alignment, Gene expression level and cluster analyses, GO analysis of the DEGs, KEGG analysis of DEGs are clear. As previously mentioned, while the novelty value of the results is reasonablethe authors must avoid the use of both the word “significant” and the “probability value itself”; it is a pleonasm. Regarding to the Discussion section, the authors initiated this section including the working hypothesis of the study. The authors defined if, with the obtained results, the working hypothesis is rejected or non-rejected. For this reason, the authors certainly included the working hypothesis prior to the objectives in the Introduction section. Excellent. The authors must followed, as muach as possible, the same order in this section according to that proposed in the Results section. The authors linked in a logical fashion their main findings along with the discussion section, compared and discussed their main findingsand. Then the authors proposed some possible explanations for such specific outcomes, considering to previous similar studies from the scientific literature. Excellent. In general, the main outcomes of the study were enhanced and well  presented in this R1 version. As previoussly mentioned, this is a very interesting study. Yet, the authors still need to improve the clarity and logical arrangement of the main outcomes, especially in the Introduction and Results sections.  All the commented issues and requests should be clearly addressed by the authors. At this point, and based on the above comments, my pronouncement is that this manuscript still requires moderate corretions prior to the final acceptance.

Author Response

Comment 1: This is the R1 (i.e., Revised 1 version) of the paper of Feng & colleagues; they aimed to investigate the molecular mechanisms of wool and skin responses to cold stress in lambs belonging to two breeds with varying cold tolerance. In general, the organization of the experiment seems to be well designed, while the sentence structure and grammar were certainly improved; nonetheless, some key suggestion previousy proposed for this Reviewer were not addressed as required. The Introduction section is still quite long and the authors instead of contextualize their study including the importance of Asia and China regarding the world sheep population and another key issues from a social, economic, and productive point of view, especially with respect to wool/fiber-producing breeds used in the study, they included, instead, phenotipic information of both breeds which does not add any scientific power to the paper. The authors must contextualize the importance of they research with a global perspective. This Introduction section must be shortened; this is a must. Most of the suggestions proposed to enhance the sections

AU: For R1 the introduction was shortened in places to meet the requirements of this reviewer (formerly reviewer #2) and lengthened to meet the requirements of Reviewer #1 (who wanted greater detail on the background and characteristics of the two breeds).

We have now made another attempt to meet the requirements of both reviewers, with this section being shortened from 801 words in length to 545 words (greater than a 30% reduction in length). We do stress that while wool and skin are a focus of part of this study, the primary focus is the response of lambs to a cold challenge, not the ‘social, economic, and productive matters’ relating to these wool producing breeds. Both the breeds studied are also used for meat production, but in the context of the response to a cold challenge, one cannot ignore the role played by the skin and the wool.

Comment 2: Material and Methods as weel as Results were well approached, but there ae some issues that must be addressed. L288-290; please rewrite, this is a non-sense sentece structure. Again, the authors do not define when there were simple effects or interaction effects. If there is NO interaction effect, meaning that a factor A does not depend on a factor B, then they will need to address the simple effects for the affected response variables. If, on the contrary, the ANOVA defines an interaction effect, it means that factor A depends on factor B. Specifically, generate a subheading for each scenario; one subeading for single effects and another subheading for interaction efects, and include the respective Tables including only the significant effects in the Results section, and the non-significant ones may be included in a different Table(s) in the supplementary information. This had already been required, but the authors did not solve it in the best way.

AU: We hope we have now met this requirement, specifically re-drafting all the text and producing two tables that illustrate the significant effects. In the context of not being perceived to have ‘cherry-picked’ ONLY the significant results, we have maintained ALL the comparison results in the Supplementary Tables.

Comment 3: The quality assessment of RNM-Seq data and genome alignment, Gene expression level and cluster analyses, GO analysis of the DEGs, KEGG analysis of DEGs are clear. As previously mentioned, while the novelty value of the results is reasonablethe authors must avoid the use of both the word “significant” and the “probability value itself”; it is a pleonasm.

AU: The word ‘'significant' (or derivatives thereof) is only used five times in the whole manuscript:

1) Line 282-284.... Upon confirming assumption compliance, post-hoc pairwise comparisons among the four groups were executed via the least significant difference (LSD) method to delineate specific subgroup disparities.

2) Line 285-287....  Significance was defined at p < 0.05, with all results reported as means ± standard error of the mean (SEM) to ensure precision in effect size interpretation.

3) Line 468-469.... Most DEGs were related to the proportion of fine wool fibres, with GNLY, specific to the Hulunbuir lambs in colder environments being the most significant.

4) Line 503-506…. It is therefore notable that in our study, the epidermis thickness of the Hulunbuir lambs was greater at -20°C compared to 15°C, while the Hu lambs also exhibited increased epidermis thickness at -20°C, but this difference was not significant.

5) Line 519-521…. Li [29] found that rectal temperature in 6-month-old Dupo and Suffolk lambs decreased after eight days of cold exposure, although the difference was not significant.

In all five iterations no P-value was given, so we are confused by this comment.

Comment 4: Regarding to the Discussion section, the authors initiated this section including the working hypothesis of the study. The authors defined if, with the obtained results, the working hypothesis is rejected or non-rejected. For this reason, the authors certainly included the working hypothesis prior to the objectives in the Introduction section. Excellent. The authors must followed, as muach as possible, the same order in this section according to that proposed in the Results section.

AU: We believe that we have, and we have checked this once again.

Comment 5: The authors linked in a logical fashion their main findings along with the discussion section, compared and discussed their main findingsand. Then the authors proposed some possible explanations for such specific outcomes, considering to previous similar studies from the scientific literature. Excellent. In general, the main outcomes of the study were enhanced and well  presented in this R1 version. As previoussly mentioned, this is a very interesting study. Yet, the authors still need to improve the clarity and logical arrangement of the main outcomes, especially in the Introduction and Results sections.  All the commented issues and requests should be clearly addressed by the authors. At this point, and based on the above comments, my pronouncement is that this manuscript still requires moderate corretions prior to the final acceptance.

AU: We have revised the manuscript accordingly by clarifying the Introduction, improving the structure of the Results, and refining transitions to enhance overall readability and logical flow.